# Hundreds of dual-stage antimalarial molecules discovered by a functional gametocyte screen

Celia Miguel-Blanco[1,2], Irene Molina[1], Ana I. Bardera[3], Beatriz Díaz[1], Laura de las Heras[1], Sonia Lozano[1], Carolina González[1], Janneth Rodrigues[1], Michael J. Delves[2], Andrea Ruecker[2,†], Gonzalo Colmenarejo[3], Sara Viera[1], María S. Martínez-Martínez[1], Esther Fernández[1], Jake Baum[2], Robert E. Sinden[2] & Esperanza Herreros[1]

*Plasmodium falciparum* stage V gametocytes are responsible for parasite transmission, and drugs targeting this stage are needed to support malaria elimination. We here screen the Tres Cantos Antimalarial Set (TCAMS) using the previously developed *P. falciparum* female gametocyte activation assay (*Pf* FGAA), which assesses stage V female gametocyte viability and functionality using *Pf*s25 expression. We identify over 400 compounds with activities <2 µM, chemically classified into 57 clusters and 33 singletons. Up to 68% of the hits are chemotypes described for the first time as late-stage gametocyte-targeting molecules. In addition, the biological profile of 90 compounds representing the chemical diversity is assessed. We confirm *in vitro* transmission-blocking activity of four of the six selected molecules belonging to three distinct scaffold clusters. Overall, this TCAMS gametocyte screen provides 276 promising antimalarial molecules with dual asexual/sexual activity, representing starting points for target identification and candidate selection.

[1] Diseases of the Developing World (DDW), GlaxoSmithKline, Madrid, Tres Cantos 28760, Spain. [2] Department of Life Sciences, Imperial College of London, London SW7 2AZ, UK. [3] Molecular Discovery Research (MDR), GlaxoSmithKline, Madrid, Tres Cantos 28760, Spain. † Present address: Mahidol-Oxford Tropical Medicine Research Unit, Faculty of Tropical Medicine, Mahidol University, Bangkok 10400, Thailand and Centre for Tropical Medicine and Global Health, Nuffield Department of Medicine, University of Oxford, Oxford OX3 7FZ, UK. Correspondence and requests for materials should be addressed to E.H. (email: eherreros.a@gmail.com).

During the last 10 years, there has been a substantial intensification of malaria control and prevention with the implementation of both new vector control strategies and chemotherapies. Nevertheless, the disease still remains a worldwide public health challenge with *Plasmodium falciparum* causing more than 214 million cases and 438,000 deaths per year[1]. Malaria elimination is now being discussed as a realistic goal, but new tools will be needed to achieve this aim, particularly in areas of high transmission[2]. It is unlikely that malaria elimination can be achieved in all contexts unless drugs and/or vaccines that interrupt malaria transmission can be discovered, developed and deployed[3]. *Plasmodium* stage V gametocytes are solely responsible for malaria transmission. Thus, they constitute an important target to block the parasite lifecycle through drug administration to infected patients or asymptomatic carriers.

Since 2001, artemisinin-based combination therapies (ACTs) have been recommended as first-line treatment for uncomplicated *falciparum* malaria[4]. Although ACTs rapidly remove asexual blood stages and early gametocytes from the patient, their effectiveness against transmissible stage V gametocytes remains unclear[5–7]. The only chemotherapeutic intervention available for transmission interruption is low-dose ($0.25\,\mathrm{mg\,kg}^{-1}$) primaquine[8–11], recommended for use in areas of low transmission as a single administration following ACT in patients with *P. falciparum* malaria[12]. Higher doses may be more efficacious, but the use of primaquine for transmission reduction is limited because of the potential for haemolysis in individuals with glucose-6-phosphate dehydrogenase (G6PD) deficiency, a relatively common genetic variation found in malaria endemic areas[13]. When considered with the recently reported cases of artemisinin resistance in the Greater Mekong subregion that are threatening the effectiveness of ACTs[14–16], the discovery of safer drugs with new modes of action for malaria treatment, prevention and transmission interruption is more urgent than ever.

Several assays have been developed in recent years with the objective of identifying drugs with transmission-blocking potential. Drugs with gametocytocidal activity have been detected using diverse readouts, such as metabolic parameters (ATP, pLDH, oxidoreduction)[17–19] or mitochondrial damage and luciferase reporters to track different gametocyte stages[20,21]. Most of these assays are amenable to high throughput screening (HTS) of large compound libraries[20,22,23]. Alternatively, a new generation of gametocyte assays assess male and/or female gamete formation as a broader metabolic framework within which to determine drug activities modulating stage V gametocyte functionality and viability. The idea was first reported by Delves and colleagues[24], and subsequently developed into a single assay, the *P. falciparum* Dual Gamete Formation Assay (*Pf* DGFA)[25]. To date, a limited but diverse group of studies[24–27] suggest that results obtained in the *Pf* DGFA usefully correlate with those of the *ex vivo* standard membrane feeding assay (SMFA), but we remain alert to the need to confirm *in vitro* activities with those *in vivo*. The *P. falciparum* female gametocyte activation assay (*Pf* FGAA) was the first to be scaled to 384-well format and validated for HTS[28].

The Tres Cantos Antimalarial Set (TCAMS) is the largest published collection of compounds active against *P. falciparum* asexual blood stages[29]. In this study, the 13.5 K compounds in the TCAMS are screened against stage V gametocytes in the *Pf* FGAA[28] to identify molecules effective against female gamete formation. As both female and male gametes are required for the development of mosquito stages, the parasite lifecycle may be interrupted by solely targeting one of them. The objective of this study is the discovery of new chemical diversity with activity against both asexual blood stages and stage V (female) gametocytes that may not only treat clinical symptoms but also block malaria transmission.

## Results

**Hit identification**. The TCAMS was screened in the *Pf* FGAA following the progression cascade described in Fig. 1. The 13,533 compounds were tested once at $2\,\mu\mathrm{M}$ single concentration using 48 h exposure, and 755 hits were identified using a mean plus 3 s.d.'s statistical cut-off, corresponding to 53% inhibition for this screen. The hits were then re-tested in triplicate at the same concentration resulting in 405 compounds with confirmed activity (3% final hit rate). These compounds were then evaluated in dose–response to determine their 50% inhibitory concentration ($\mathrm{IC}_{50}$) (Supplementary Data 1). A good correlation was found between the confirmed hits and dose–response assay with more than 80% of compounds possessing an $\mathrm{IC}_{50} < 2\,\mu\mathrm{M}$. Additionally, the cytotoxicity of these molecules was evaluated in mammalian cells (HepG2) to determine their specificity for the parasite. Up to 120 compounds showed 50% inhibitory concentration ($\mathrm{Tox}_{50}$) values above $10\,\mu\mathrm{M}$ (Supplementary Data 1).

In parallel, the set of confirmed hits from the *Pf* FGAA screen was cross-compared in the gametocyte ATP-depletion assay[17], at $2\,\mu\mathrm{M}$ single concentration (Supplementary Data 1). This assay monitors ATP levels as surrogate of gametocyte viability. We observed that of the 405 compounds which were identified by the

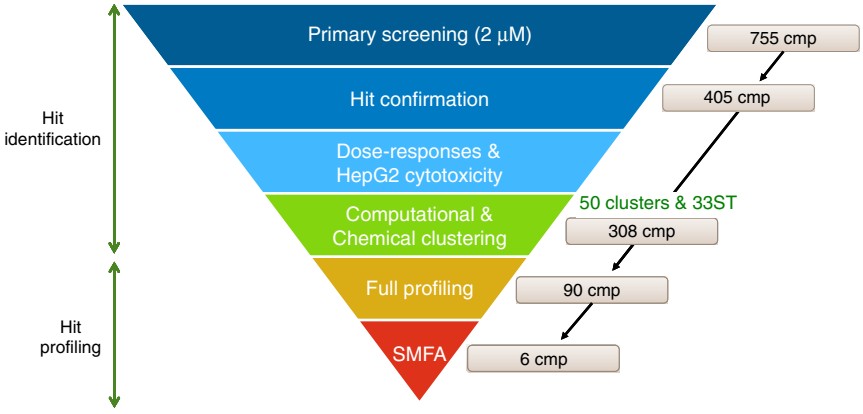

**Figure 1 | Progression cascade of the TCAMS screen in the *Pf* FGAA.** The different steps followed during the TCAMS screen as well as the number of compounds identified in the first phases and those selected for further profiling are shown. The first four steps can be defined as 'hit identification'. After clustering, compounds are progressed to characterize their biological profiles and finally tested in the gold standard SMFA.

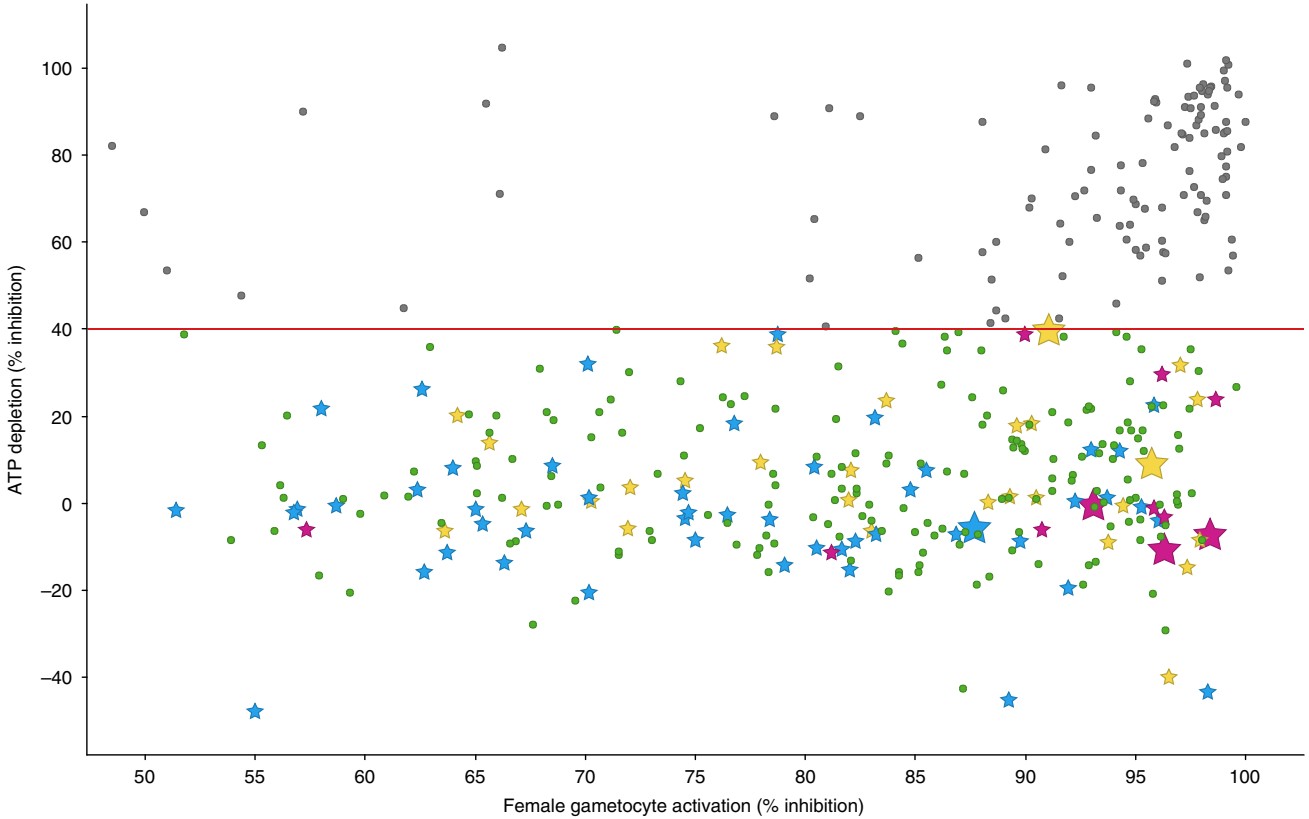

**Figure 2 | Correlation graph of hits identified in the *Pf* FGAA screen compared to their activities in the ATP-depletion assay.** 276 compounds (green circles and all stars) out of the 405 hits were exclusively identified in the *Pf* FGAA while 129 compounds (grey circles) were active in both assays. Stars represent those compounds further profiled: slow-acting (blue stars), fast-acting (yellow starts) and gamete-targeting (magenta stars). Compounds tested in the SMFA are marked as big stars.

*Pf* FGAA, only 129 compounds were also identified in the ATP-depletion assay (40% inhibition cut-off) (Fig. 2).

To assess the chemical diversity provided by the *Pf* FGAA screen, all confirmed hits (405 compounds) were subjected to computational analysis by applying a complete-linkage clustering algorithm[30], using a Tanimoto similarity threshold of 0.55. This analysis identified 82 compound clusters and 63 singletons. After chemical reviewing, they were reorganized based on structure similarity into 57 clusters and 33 singletons. Initial triage was based on simple and relaxed physicochemical properties (molecular weight, lipophilicity, number of aromatic rings) and selectivity index. Seven clusters comprising 97 compounds with molecular weights above 600 g mol$^{-1}$, clogP above 6 and HepG2 cytotoxicity $< 10 \mu M$ were flagged and deprioritized from further biological profiling. This left a total of 308 compounds classified in 50 clusters, plus 33 singletons for consideration (Supplementary Data 2). Clusters with three or less representatives comprised more than 70% of the set, which is indicative of the large chemical diversity identified in this screen.

A subset of 90 compounds representative of the new chemical diversity was selected for further characterization of the parasitological profile (Supplementary Data 3). Some of these chemical series are depicted in Fig. 3 and Supplementary Fig. 1. Compound selection was performed taking into consideration the top potency (IC$_{50}$) and cytotoxicity (Tox$_{50}$) values of compounds in each cluster.

**Hit profiling**. The parasitological properties of the selected molecules were further investigated to assess their speed of action

and activity against female gametocytes or gametes (Fig. 4 and Supplementary Data 3).

To determine their speed of action, gametocytes were preincubated with the compounds for only 24 h before inducing gamete formation and their IC$_{50}$ values compared to the ones obtained in the 48 h preincubation described above (Supplementary Data 3). There were 49 compounds that were only active with a 48 h exposure (designated as slow-acting), while 29 compounds additionally showed activity with only 24 h exposure (designated as fast-acting) (Fig. 4).

Moreover, when the compounds were added 30 min after triggering gametocyte activation, 12 compounds prevented cell surface expression of *Pfs*25 (designated as gamete-targeting; Fig. 4). This suggests that these molecules target female gametes directly, although this does not discount the possibility that they are also active against stage V gametocytes.

**Validation of the *in vitro* transmission-blocking activity.** To validate the transmission-blocking potential of the TCAMS hits and assess the predictive value of the *Pf* FGAA as an *in vitro* high throughput surrogate of full mosquito feeding, the 'gold-standard' SMFA was performed. Six compounds showing different biological profiles, belonging to four distinct chemical clusters, were selected based on their potencies in the *Pf* FGAA (Supplementary Data 1, Table 1). For each compound, inhibition of exflagellation of male gametocytes was measured. Five compounds showed more than 50% inhibition while TCMDC-124559 displayed 30% inhibition (Fig. 5, Table 1). SMFAs were then performed with a single dose at a concentration equivalent to the respective *Pf* FGAA 90% inhibitory concentration (IC$_{90}$) at the

**Figure 3 | TCAMS chemical series representatives of the compound set further progressed.** Scaffolds of 15 chemotypes, belonging to 24 of the 90 compounds profiled, that are within an appropriate physicochemical space (clogP < 5, number of aromatic rings < 3).

48 h incubation time (Table 1). Compounds were tested in duplicate in 2–3 independent experiments in the indirect format of the assay, that is, gametocytes were exposed to the drug for 48 h before mosquito feeding to replicate a drug exposure time equivalent to that used in the *Pf* FGAA. Five of these selected compounds showed a reduction in oocyst prevalence of 58–100% (Fig. 6a), and a reduction in oocyst intensity of > 80% (Fig. 6b).

By contrast, TCMDC-124559 reduced prevalence and intensity by < 20% (Fig. 6a,b, respectively).

***In vivo* efficacy and pharmacokinetics of selected molecules.** Two out of the six compounds progressed to SMFA, TCMDC-123767 (cluster 30) and TCMDC-141154 (cluster 4), were

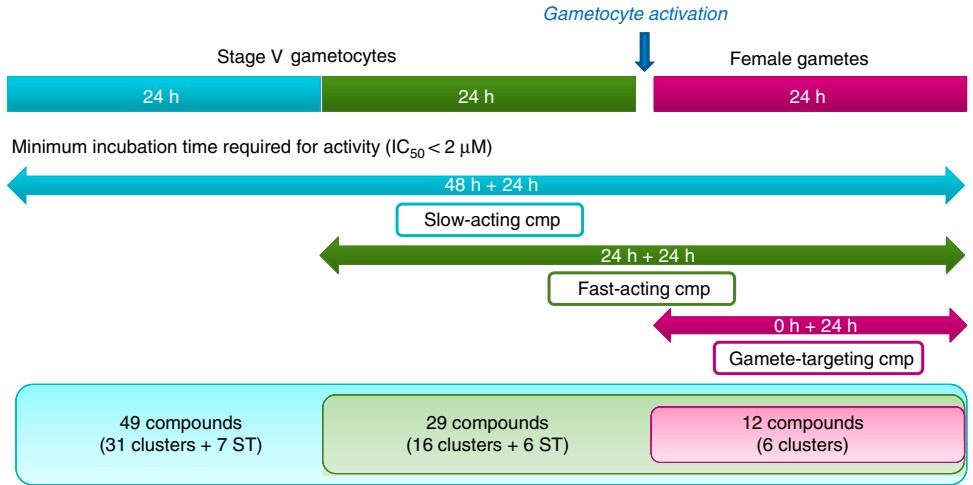

**Figure 4 | Diversity of biological profiles determined by use of different incubation times in the *Pf* FGAA.** Stage V gametocytes were exposed to the 90 selected compounds for 24 or 48 h to determine the drug inhibitory speed of action. In addition, compounds were added after triggering gametocyte activation to evaluate their activity in female gametes. The figure shows the number of compounds and chemical diversity classified under each category, that is, slow-acting, fast-acting or gamete-targeting compounds.

evaluated *in vivo* in the *Plasmodium berghei* murine model[31]. Selection criteria was based on the compound potencies in both asexual stages and gametocytes in the *Pf* FGAA, as well as on their transmission-blocking activity observed in the SMFA (Table 1). Owing to compound availability, TCMDC-141154 had to be replaced by a close analogue from cluster 4: TCMDC-140549 (Supplementary Data 3). Compounds were tested at 50 mg kg$^{-1}$ single dose 2-day oral treatment and efficacy was measured as a reduction of asexual blood stages parasitaemia greater than 40% compared with the vehicle-treated mice. *P. berghei* growth was not reduced above the threshold after mice treatment with any of the two compounds (Fig. 7).

Additional *in vitro* ADME and *in vivo* pharmacokinetic studies were then performed to explain the lack of activity of these two compounds. TCMDC-123767 was rapidly eliminated *in vivo* and was last detected in blood above the lower limit of quantification (LLOQ, 2.5 ng ml$^{-1}$) only 1 h post-administration. Supporting this, it also showed high *in vitro* microsome clearance (7.9 ml min$^{-1}$ per g liver). TCMDC-140549 was estimated to possess moderate permeability through artificial membranes (170 nm s$^{-1}$), which may drive a low oral bioavailability.

## Discussion
To the best of our knowledge, this paper reports the largest screening effort performed to date using an assay that utilizes *P. falciparum* gamete formation as the endpoint. The TCAMS library was screened in the *Pf* FGAA and 405 chemical-starting-points for antimalarial transmission-blocking drug discovery were identified. Further analysis of these molecules revealed that 276 compounds (25% with selectivity index >10) belonged to chemotypes never described before as gametocyte-targeting molecules.

Regarding the chemical diversity identified, a few of these compounds showed similarity with classical antimalarial scaffolds, for example, 4-aminoquinoline (TCMDC-138933, cluster 7, IC$_{50}$ = 0.76 μM) and diaminopyrimidine (TCMDC-137820, cluster 25, IC$_{50}$ = 1.06 μM) (Supplementary Data 3). The majority comprises novel chemotypes with good physicochemical properties. This TCAMS set of hits includes several scaffolds that have

been previously identified by GSK as promising antimalarial starting points for drugs against parasite asexual stages[32], and that are or have been part of internal medicinal chemistry programmes (Supplementary Fig. 1). These potential drugs can now be prioritized with the added value of being gametocyte-targeting. Interestingly, a preliminary analysis of hit structure revealed that some of the compounds are closely related to the GSK Published Kinase Inhibitor Set (PKIS)[33] (Supplementary Fig. 1), suggesting *Plasmodium* kinases as potential targets for these molecules.

The original literature annotation for TCAMS was also investigated. TCMDC-141611 is a compound patented by SmithKline Beecham as an inhibitor of the Tie-2 tyrosine kinase receptor, which is involved in angiogenesis. Despite the lack of a classical tyrosine kinase family in the *P. falciparum* kinome[34], there is evidence for tyrosine phosphorylation being involved in regulatory functions in the parasite[35]. TCMDC-142257 is an antagonist of the Dopamine 2 receptor, a kind of G-protein-coupled receptor, for which there is some bioinformatic evidence in the *P. falciparum* genome[36]. Further investigations will be necessary to understand the mechanisms of action of these compounds and by publishing these structures we intend to stimulate research in this area.

From the point of view of malaria drug discovery, the most practical way to interrupt the *Plasmodium* lifecycle would be through the clearance of transmissible stages present in the peripheral bloodstream of infected patients. In this context, those compounds with a fast-acting profile would be preferred to those requiring longer exposure times. Overall, almost 90% of the compounds tested in this study exerted their effect in stage V gametocytes (and asexual blood stages) but not in female gametes, which revealed their potential utility for transmission-blocking strategies. Further research would be required to determine if the 12 compounds active against female gametes also target stage V gametocytes and so might have an added value as multi-stage antimalarial drugs.

Six selected compounds with diverse biological and chemical profiles were tested in the SMFA (Table 1). Four out of the six tested compounds showed more than 80% block in transmission (Fig. 6, Table 1), while TCMDC-124559 and TCMDC-125345 had

**Table 1 | Biological and chemical profile of the six compounds tested in the SMFA.**

| TCAMS ID | Chemical structure | Cluster number | *Pf*FGAA IC$_{50}$ (µM) | SMFA concentration (µM) | Inhibition of exflagellation (%) | Block in transmission (%) | Tox$_{50}$ (µM) | Biological profile |
|---|---|---|---|---|---|---|---|---|
| TCMDC-123767 | | 30 | 0.16 | 1 | 88 | 83 | 100 | Slow-acting |
| TCMDC-125345 | | 18 | 0.36 | 1 | 50 | 60 | 39.82 | Fast-acting |
| TCMDC-141698 | | 11 | 0.44 | 1 | 84 | 82 | 4.79 | Gamete-targeting |
| TCMDC-141070 | | 4 | 0.53 | 2 | 99 | 88 | 28.96 | Gamete-targeting |
| TCMDC-141154 | | 4 | 0.21 | 1 | 97 | 93 | 11.9 | Gamete-targeting |
| TCMDC-124559 | | 18 | 0.5 | 1 | 30 | 15 | >100 | Fast-acting |

lower efficacy, even though all compounds had IC$_{50}$ values between 0.16 and 0.5 µM in the *Pf* FGAA. However, there was a good correlation between inhibition of exflagellation, reduction in oocyst intensity and block in transmission (Figs 5 and 6). This suggests that compounds affecting both male and female gamete formation may lead to a more efficacious blockade of malaria transmission.

Further *in vivo* evaluation of two selected molecules, TCMDC-123767 and TCMDC-140549, in the *P. berghei* murine model[31] showed an ED$_{50}$ > 50 mg kg$^{-1}$ after a single dose 2-day treatment (Fig. 7). Subsequent pharmacokinetic analysis revealed very low exposure levels in blood for TCMDC-123767, which might explain the lack of effect of this molecule in the *in vivo* model. Given that both compounds showed a good

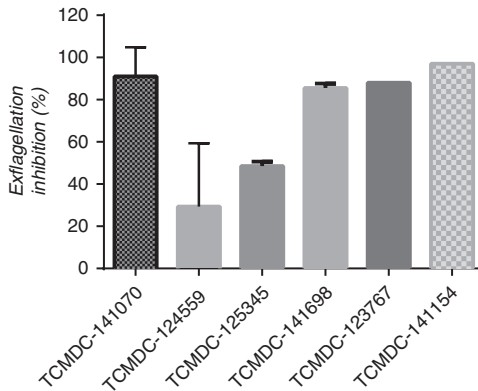

**Figure 5 | Exflagellation inhibition by six TCAMS compounds.** Stage V gametocyte cultures were exposed to 2 μM of TCMDC-141070 and 1 μM of the five remaining compounds for 48 h. The percentage inhibition was determined compared to DMSO-treated controls. Compounds were then progressed into SMFA to further determine the transmission blocking in mosquitoes. Each bar represents mean value of two (TCMDC-125345, TCMDC-141698, TCMDC-123767, TCMDC-141154) or three (TCMDC-141070, TCMDC-124559) independent replicates with s.d.

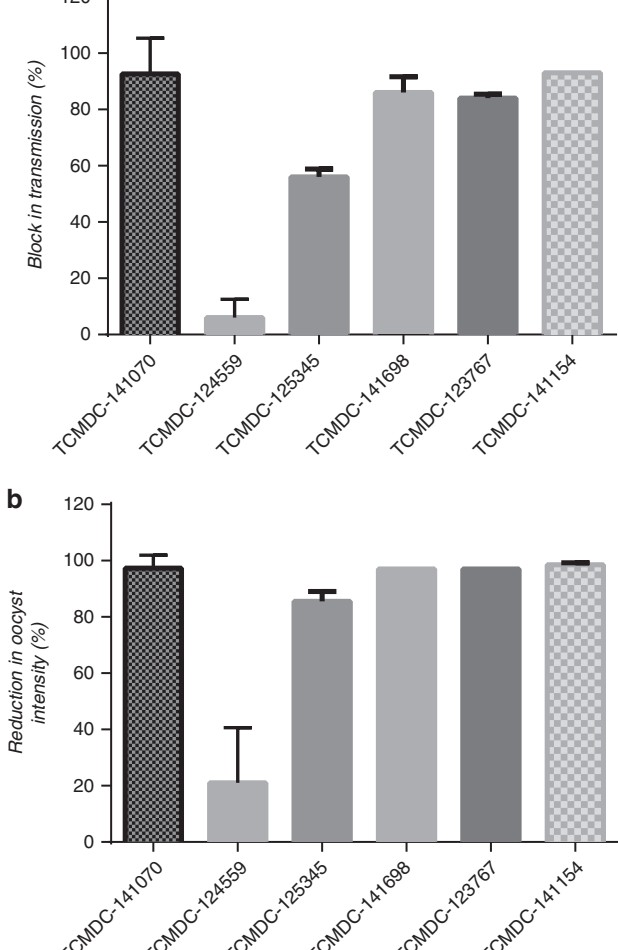

**Figure 6 | Effect of selected TCAMS compounds in the SMFA.** The following parameters were measured: (**a**) Transmission-blocking activity (prevalence reduction) and (**b**) reduction in mean *P. falciparum* oocyst intensity. Each bar represents mean value of two (TCMDC-125345, TCMDC-141698, TCMDC-123767, TCMDC-141154) or three (TCMDC-141070, TCMDC-124559) independent replicates with s.d.

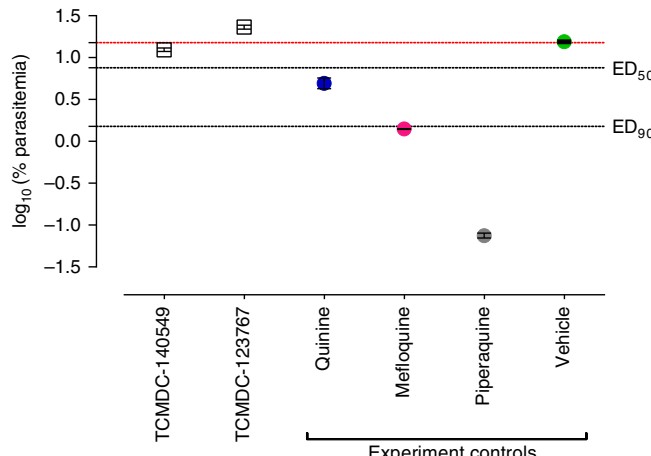

**Figure 7 | Effect of two TCAMS hits in the *P. berghei in vivo* mouse model.** Each experiment included a control group treated with vehicle (green circle) used as a reference to calculate the percentage of inhibition of parasitaemia in peripheral blood (dotted red line). The response of standard antimalarials in the same assay is also presented (blue, magenta and grey circles). Represented data are the mean ± s.e.m. $\log_{10}$ [% asexual blood stages at day 4] of two mice per point.

parasitological profile *in vitro*, further studies to improve stability and bioavailability will be performed to address their potential use as antimalarial drugs. Clearance of TCMDC-123767 and TCMDC-140549 may be reduced by blockade of labile positions, whereas solubility of TCMDC-123767 may be increased by incorporation of polar and/or ionizable groups.

In conclusion, the *Pf* FGAA can identify potential transmission-blocking chemotypes which, due to the biological content covered in this assay, were undetected by previous gametocyte-based assays. The TCAMS library screen yielded 276 new gametocyte-targeting molecules belonging to 57 diverse chemical entities and 3 biological profiles. Nearly 25% of the hits identified are within the appropriate physicochemical space (clogP < 5, number of aromatic rings < 3) and will therefore be considered for further progression, while other hits represent valuable starting points as tool compounds to identify new modes of action involved in the biology of *Plasmodium* transmission.

## Methods

**Gametocyte production.** *P. falciparum* NF54 (originally isolated from an imported malaria case in the Netherlands in the 1980s; BEI Resources, cat. no. MRA-1000) asexual-stage culture was performed as described by Trager and Jensen[37]. Cultures synchronized at the ring stage were used to start gametocyte cultures (day 0) at 1% parasitaemia and 4% haematocrit in 200 ml final volume using culture bottles (Nalgene 3110-42, Thermo Scientific). Complete culture medium (RPMI 1640 supplemented with 25 mM HEPES, 50 μg ml$^{-1}$ hypoxanthine, 2 g l$^{-1}$ NaHCO$_3$ and 10% pooled human male type A+ serum) was totally replaced daily for 14 days without fresh erythrocyte addition. Human serum was obtained from Interstate Blood Bank, A+ serotype; no aspirin 2 h before drawing and no antimalarials 2 weeks before drawing. To ensure a stable temperature at 37 °C, which is crucial for gametocyte production and maturation, pre-warmed medium and a slide warmer (XH-2001, Premiere) were used. Under these conditions, parasitaemia reaches a peak of asexual stages on days 4–5, and the first gametocytes are differentiated in culture on day 6. Sexual-stage development was monitored microscopically by Giemsa-stained thin blood smears at day 7 (mainly asexual stages and stages I to III gametocytes) and day 14 (stages IV and V gametocytes).

**Female gametocyte activation assay (*Pf* FGAA).** Cultures showing mainly stage V gametocytes were purified by differential sedimentation as previously reported[28]. The gametocyte concentration was adjusted to plate 8,000 gametocytes per well (100 μl per well) in 384-well poly-D-lysine coated plates (781946, Greiner Bio-One) containing the compounds to test. Gametocytes were incubated with drugs for

either 24 h or 48 h at 37 °C (3% $O_2$, 5% $CO_2$ and 92% $N_2$). Female gametocyte activation was then triggered by temperature drop and addition of xanthurenic acid (D120804, Sigma) as previously described[24]. To detect female gametes, monoclonal anti-Pfs25 antibody[38] 4B7 (BEI Resources (formerly MR4), cat. no. MRA-315) conjugated to the Cy3 fluorochrome (GE Healthcare) was added to ookinete medium (RPMI 1640 supplemented with 25 mM HEPES, 50 $\mu$g ml$^{-1}$ hypoxanthine, 2 g l$^{-1}$ NaHCO$_3$, 100 $\mu$M xanthurenic acid and 20% human serum) at a final concentration of 0.5 $\mu$g ml$^{-1}$. Activated cultures were then kept at 26 °C for 24 h protected from light, and image acquisition was performed in the Opera High-Content Screening System (PerkinElmer). Using $\times$ 10 air objective, five images per well were taken using 532 nm excitation. Images were analysed with a Columbus image data storage and analysis system (PerkinElmer) based on a script specifically designed for this assay that considers size, roundness and intensity of fluorescence of the female activated gametocytes.

**Gametocytocidal ATP-depletion assay.** 'Viability' of mature gametocytes was determined using the gametocyte ATP bioluminiscence assay[22]. Briefly, stages IV–V gametocyte cultures were double-purified by differential sedimentation followed by magnetic isolation[17]. Then, parasites (50 $\mu$l per well containing 12,500 gametocytes) were added to compound pre-dispensed 384-well plates (781091, Greiner Bio-One) and incubated at 37 °C for 48 h (3% $O_2$, 5% $CO_2$ and 92% $N_2$). BacTiter-Glo kit (G8231, Promega) was used to determine the ATP levels of 'live' parasites according to the manufacturer's instructions. After reagent addition (50 $\mu$l per well), luminescence of the plates was measured using a microplate reader (HTS counter Victor, Wallac).

**Exflagellation assay.** Exflagellation was measured as previously reported[24] with minor modifications. Using 5 ml of day-14 gametocyte cultures, parasites were incubated with the corresponding concentration for each compound in six-well plates at 37 °C (3% $O_2$, 5% $CO_2$ and 92% $N_2$). After 48 h incubation, a 200 $\mu$l sample was spun down in a microfuge, the supernatant was carefully removed and the pellet resuspended in 15 $\mu$l of pre-warmed ookinete medium. Parasites were placed in Fast-Read disposable haemocytometer slides and, after 15 min at room temperature (22 °C), exflagellation centres per field were counted under $\times$ 10 objective.

**Standard membrane feeding assay (SMFA).** Day-14 gametocyte cultures were plated in pre-warmed six-well-plates (5 ml per well) and were exposed to the compounds (1 $\mu$M or 2 $\mu$M final concentration) for 48 h at 37 °C (3% $O_2$, 5% $CO_2$ and 92% $N_2$). Within this incubation time, 3 ml of media were removed after the first 24 h and replenished with the same volume of fresh media with compound added to obtain the required final concentration. After 48 h of total exposure, cultures were centrifuged at 2,500 $g$ for 3 min at 37 °C, diluted 1:1 with 100% packed volume of fresh erythrocytes and finally formulated as artificial mosquito blood meals at 50% haematocrit with pre-warmed human serum. All steps were performed at 37 °C to avoid gametocyte activation. Four- to six-day old female Anopheles stephensi mosquitoes (GlaxoSmithKline Insectary) were fed for 30 min via Parafilm membrane attached to glass feeders (12831283, Fisher Scientific) connected to a 37 °C circulating water bath. Fed mosquitoes were maintained in an incubator at 27 °C and 75% relative humidity with 12 h light/dark cycles. Seven to eight days post-feeding, mosquitoes with fully developed ovaries were dissected for midguts, which were rinsed in a 0.2% mercury-dibromofluorescein (63869, Fluka) in water solution for 10 min. Total number of oocysts in individual midguts were counted using a light microscope (DM2000, Leica) under a $\times$ 10 objective. The percentage of block in transmission (reduction in prevalence) and the percentage of reduction in mean oocyst intensity were calculated after normalizing to the control DMSO-treated sample. Two to three independent SMFA experiments were performed and two internal duplicates of 40 mosquitoes each were used per compound. Mean oocyst intensities were not less than two oocysts per mosquito and the prevalence of infection ranged from 70 to 90% in the fed mosquito control groups. See complete experimental data set in Supplementary Table 1 and Supplementary Note 1.

**HepG2 cytotoxicity assay.** Actively growing HepG2 cells (HB-8065, ATCC) were detached from the culture surface and dispersed with 5 ml of Eagle's Minimum Essential Media (supplemented with 10% FBS/1% NEAA solution/1% penicillin + streptomycin) by repeated pipetting. Cell suspension was added to 500 ml of the same medium at a final density of $1.2 \times 10^5$ cells ml$^{-1}$ and 25 $\mu$l per well were seeded in 384-well-plates with pre-dispensed compounds (250 nl per well) using a Multidrop combi dispenser (Thermo Scientific); this number of cells (typically 3,000 cells per well) ensures that new monolayers were not more than $\sim$50% confluent at the time of seeding. Cells were incubated at 37 °C and 5% $CO_2$ in a humidified incubator for 48 h. After incubation, plates and CellTiter-Glo Reagent (G7571, Promega) were equilibrated at room temperature for 30 min before proceeding to develop the luminescent signal. Using a Multidrop combi dispenser, 25 $\mu$l per well of the signal developer were added to the plates and after 10 min at room temperature for stabilization, plates were read on the ViewLux system (Perkin Elmer).

**Evaluation of in vivo antimalarial therapeutic efficacy.** Pathogen-free CD1 mice (Hsd:ICR) were obtained from Harlan Interfauna Iberica (Barcelona, Spain). Eight weeks old female CD-1 mice were infected intravenously with $10^7$ infected erythrocytes (day 0). Dosing solutions were prepared in water containing 5% DMSO/20% Captisol, at a target dose of 50 mg kg$^{-1}$. Mice received two oral doses once a day according to their body weight (20 ml kg$^{-1}$) starting at day 2 after infection. Control mice received the same treatment schedule with the vehicle used for the compound preparation. Samples from mice peripheral blood were taken before starting treatment and 24 h after finishing the dosing, to measure parasitaemia by flow cytometry, using the YOYO-1 staining[39], to assess the reduction of parasitaemia compared with the vehicle-treated mice.

**In vivo pharmacokinetic studies.** Female CD-1 mice (Harlan Interfauna Iberica) at 8 weeks of age were used for single oral dose pharmacokinetic studies ($n = 2$). Dosing solutions were prepared in 20% (v:v) Captisol in water, at a target dose of 50 mg kg$^{-1}$ (dose volume of 20 ml kg$^{-1}$). After oral dosing, blood samples (25 $\mu$l) were collected at 30 min, 1 h, 6 h and 8 h post dose for TCMDC-123767. All the blood samples were diluted with 25 $\mu$l of an aqueous solution of saponine 1% (w:v), and stored at $-80$ °C until analysis. Mice blood samples were analysed for each compound using a method upon protein precipitation followed by LC-MS/MS analysis (Applied Biosystems). Data analysis of the concentration time profiles was performed by noncompartmental methods by using WinNonLin Phoenix Version 6.3.

**Intrinsic clearance assay.** Intrinsic clearance (CLi) values were determined in mouse liver microsomes (M1000, XenoTech). Test compounds (final concentration 0.5 $\mu$M) were incubated at 37 °C for 45 min in 50 mM potassium phosphate buffer (pH 7.4) containing 0.5 mg microsomal protein per ml. The reaction was started by addition of cofactor NADPH (N1630, Sigma) at 1 mM final concentration. The final concentration of organic solvent (DMSO) was limited to 0.25% of the final volume. At 0, 5, 15, 30, and 45 min, an aliquot (100 $\mu$l) was taken, quenched with acetonitrile containing an appropriate internal standard, and analysed by HPLC-MS/MS (Applied Biosystems). CLi was determined from the first-order elimination constant by nonlinear regression, corrected for the volume of the incubation and assuming 48 microsomal mouse protein per g liver. Values for CLi were expressed as ml min$^{-1}$ per g liver.

**Artificial membrane permeability assay (AMPA).** It is a 96-well plate-based assay that measures the speed of permeation of a compound (at 10 $\mu$M concentration) through a phospholipid membrane, consisting in 1.8% egg L-a-phosphatidylcholine (830051, Avanti Polar Lipids) and 1% cholesterol (C8667, Sigma) dissolved in n-decane (D0011, TCI American). Phosphate buffer (50 mM Na$_2$HPO$_4$ with 0.5% 2-hydroxypropyl-b-cyclodextrin), pH 7.05, is added to the top and bottom of the plate (S5EJ046I08, MilliPore Corp.). The lipids are allowed to form bilayers across the small holes in the filter and compound concentration is measured 3 h later in both donor and the acceptor compartments by HPLC (Agilent 1100 LC). Permeability ($P$) (nm s$^{-1}$) is calculated using the following formulas:

$$P = -10^7 \ln\left(1 - \frac{C_R}{C_{EQ}}\right) \frac{V_D}{1-X} \frac{1}{At},$$

$$C_{EQ} = \frac{(C_R V_R) + (C_R V_D)}{V_R + V_D},$$

where $X = V_D/V_R$; $C_R$ and $C_D$ are final concentrations in receiver and donor side, respectively; $V_R$ and $V_D$ are volumes in receiver and donor compartment, respectively; $A$ is the area (cm$^2$) and $t$ is the incubation time (s). The considered cut-off values are as follows: High: $P > 200$ nm s$^{-1}$; Medium: 10 nm s$^{-1}$ $< P < 200$ nm s$^{-1}$; Low: $P < 10$ nm s$^{-1}$.

**Ethics statement.** All the experiments were ethically reviewed and approved by the GlaxoSmithKline Diseases of the Developing World (DDW) Group Ethical Committee on Animal Research and were conducted according to Spanish legislation, European Directive 2010/63/EU and GlaxoSmithKline policy on the Care, Welfare and Treatment of Laboratory animals.

**Compounds and controls.** Compounds were dissolved in 100% DMSO and dispensed in the 384-well plates using an Echo-CRS liquid handler. Compounds were tested in 2–3 independent experiments. For each assay, both positive (a drug highly effective against the biological process) and negative (the vehicle consisting in 0.5% DMSO) controls were added and a minimum of ten wells per plate were analysed. Based on that, Z' factor was calculated for each plate and only those ones with values higher than 0.4 were considered acceptable for data analysis.

**Data analysis.** Data were normalized to percentage of inhibition of the biological response by using positive (that is, highest inhibition achieved by a drug, ICtrl2) or negative (that is, lowest inhibition achieved in the absence of drug but in presence

of the vehicle 0.5% DMSO, ICtrl1) controls following the equation described below:

$$\% \text{ Inhibition} = 100 - \left[ \frac{X - \text{Ctrl2}}{\text{Ctrl1} - \text{Ctrl2}} \times 100 \right],$$

where $X$ is the inhibition of measured process for the compound $X$. Ctrl1 and Ctrl2 are calculated as the average of replicates in the same microtiter plate where compound $X$ is tested.

Assay performance statistics, such as signal to background ratio, $Z'$ and robust 3 s.d. activity cut-off were calculated using templates in ActivityBase XE (IDBS, Guilford, Surrey, UK). Hit population analysis and visualization were conducted using Spotfire DecisionSite (Spotfire, Inc., Somerville, MA, USA). The $pIC_{50}$ ($- \log IC_{50}$) values were obtained using the ActivityBase XE nonlinear regression function in the full curve analysis bundle.

**Data availability.** The GSK TCAMS dataset for *P. falciparum* whole cell screening was deposited in ChEMBL-NTD, www.ebi.ac.uk/chemblntd. Additional chemical structures of compounds described in this study can be found in https://www.ebi.ac.uk/chembl/index.php/

The authors declare that all relevant data supporting the findings of this study are available within the article and its Supplementary Information files or are available from the authors on request.

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

## Acknowledgements

This work was supported by Bill and Melinda Gates Foundation grant OPP1043501.

We thank Richard Priest from Biological Sciences at GSK (Stevenage, UK) for the production and conjugation of anti-*Pfs*25 antibody. We also thank the GSK Sample Management Technologies team for the preparation of compound pre-dispensed 384-well plates. Likewise, we are grateful to Dr David Calvo and Juliana Sanchez for the culture and maintenance of *An. stephensi* colonies used to perform SMFA experiments. Naomi Richardson of Magenta Communications Ltd provided editorial assistance with the manuscript, funded by GlaxoSmithKline. Dr Jake Baum is supported by an Investigator Award from the Wellcome Trust (100993/Z/13/Z).

## Author contributions

C.M.-B., I.M. and E.H. planned, designed and supervised the work. C.M.-B. and I.M. performed the screening assays, biological profiling and contributed to data analysis. A.I.B. performed data analysis of primary screening, dose-responses and further profiling. G.C. performed the cheminformatic analysis after each screening step. B.D., L.d.l.H. and E.F. performed the computational and chemical clustering. S.L., C.G. and J.R. performed the SMFA experiments and data analysis. A.R. and M.J.D. contributed to define the different biological profiles of hits. S.V. performed the *in vivo* experiments. M.S.M.-M. performed pharmacokinetic studies. E.H., R.E.S. and J.B. managed the project. C.M.-B., I.M., B.D., L.d.l.H., J.R., S.V., M.S.M.-M. and G.C. wrote different sections of the manuscript and/or prepared figures and tables. C.M.-B. integrated individual contributions and issued the final manuscript. All authors reviewed the manuscript and accept responsibility for its publication.

## Additional information

**Competing interests:** The authors declare no competing financial interests.

**Publisher's note:** Nature remains neutral with regard to jurisdictional claims in published maps and institutional affiliations.

