## [Peer Review File · Nature Communications]

Editorial Note: Parts of this peer review file have been redacted as indicated for copyright reasons.

Reviewers' comments:

Reviewer #1 (Remarks to the Author):

There is an urgent need for additional antimalarials against all stages of the parasite, including the transmission stages and this manuscript identifies 3 scaffolds that when incubated in vitro at micromolar concentrations with late stage sexual stages reduce transmission in a standard membrane feed. This is encouraging, but statistics and the actual numbers of oocysts and infected mosquitoes need to be included to understand the significance of the SMFA experiment. For these compounds the term "transmission blocking" also needs to be changed to "transmission reducing" throughout the text. In addition for publication in Nature Communications they need to test for in vivo efficacy using the rodent malaria model or indicate that the compounds were toxic to mice and could not be tested. This in vivo data is needed to get some sense of how much additional drug development is needed to advance these compounds and must be included. To direct further screening efforts, they should also directly compare SMFA results from randomly selected compounds shown in figure 2 to inhibit gametocyte viability and those that do not. Gametocyte assays are not easy and it is important to know if it would be more efficient to screen first for viability or Pf FGAA. Their finding that only 4 of the 6 selected compounds from a total of 405 identified with the Pf FGAA assay alone may reduce transmission at micromolar concentrations is not a strong endorsement of their Pf FGAA only screening strategy.

Line 69 & 228: Please remove "increased" and revise the sentence. These assays focus on a distinct stage, but this does not mean that they have more biological content than previously used gametocyte screens that monitored ATP levels or mitochondrial potential or redox activity or luciferase expression.

Line 70: As described above data needs to be presented to state that "the in vitro gamete formation assays best approximates the activity of drugs in the mosquito stages, as determined by the ex vivo assay." No reference or data is given for this statement.

Line 140: Which 12 compounds blocked Pfs25 expression when administered after the emergence stim? Please include this in a chart and indicate them on fig 2, perhaps by circling them or using a different color star.

Line 147: In figure 2 please identify the final 6 compounds evaluated in detail as described above. What was the effect of these compounds on viability?

Line 147: More specifics are needed about how the 6 compounds were selected. Were other compounds tested and not found to be effective in the SMFA? Those selected did not include all the most potent compounds and only 4 clusters were tested, so diversity and potency were not the defining priorities. Clusters 3, 13, 26, 57 and singleton 1 all had equally potent compounds.

Line 148: Please replace "a diversity of" with "4 distinct" chemical clusters were represented.

Line 160: The number of oocysts and infected mosquitoes observed in the control and after compound treatment need to be included. If the mean for the control was 2 oocysts in 70% of the mosquitoes, which are the minimums indicated in the methods (line 316), it is difficult to know the significance of 93% inhibition. Statistics are needed to confirm that these compounds significantly reduce transmission. The concentrations used for the SMFA need to be included in Table 2.

Lines 173 & 174: Please add the cluster numbers for these two scaffolds and their EC50s in the text. Were these compounds tested in the time course? Was there a reason they were not tested in the SMFA assay?

Line 204: Please clarify this sentence. I thought all the compounds were effective against asexual parasites. At the least the compounds listed in table 1 have activity against asexual parasites. Is the sentence referring to the lack of an effect for most of the compounds in the viability assay? Although interesting from a biological perspective, from the drug development side it is a concern that the selected compounds do not affect gametocyte viability (Fig 2). What stages are not affected by the compounds? Will drug levels have to be maintained until all the sequestered stage I-IV gametocytes mature to stage V? The discussion needs to more clearly address this issue.

Line 225: Please include more discussion about why activity against females is not sufficient to predict transmission-blocking activity and how this impacts utility of this type of screen.

Reviewer #2 (Remarks to the Author):

This is concise, clear, and focused manuscript, easy to read. Authors describe logical and efficient step-wise approach to identify compounds inhibiting development of gametocytes, provide 6 well-characterized in vitro and ex vivo inhibitory compounds, and a long list of active compounds for future characterization. The manuscript will be interesting to malaria researchers and identified compounds are a good starting point in the development of transmission blocking drugs.

Nevertheless, this reviewer has a few minor suggestions/questions.

1. Authors should state in Figures 5 and 6 what their error bars are (I assume these are Standard Deviations).
2. This reviewer did not understand why only compounds with <40% inhibition in ATP assay went to further biological characterization.
3. Figure 4 states specific numbers for the inhibitory compounds in each group (slow, fast, and gamete). If possible, it would be nice to indicate in the list of 94 compounds which of them belong to which group. Also, in the scheme it says that "Minimum incubation time required for activity (IC₅₀<2μM)". I guess, it means that all compounds with IC₅₀<2 μM (measured previously at 48 hours incubation) were tested in this experiment using 24 and 0

time incubation but at what concentration? Was it a single 2 μM concentration used and efficiency of inhibition determined by the appearance of gamete? In the Supplementary table 3 IC₅₀ values for 24 and 0 time points are shown – it means that dose-response data were also obtained at these times, but this is not described in the manuscript. This reviewer counted in the Supplementary Table 3 twelve compounds with IC₅₀ < 3 μM in PfGAA at time “0” belonging to 6 groups. Are these the compounds that were active in gamete-targeting assay (two compounds indicated in the Fig 3 gamete targeting group are among these 12 that reviewer has found)? If so, it would be interesting to discuss that they might be active at time “0” because they have the lowest IC₅₀ at time “0” and the experiment in this setup provides specific IC₅₀ threshold for selection of active compounds. In any case, a better description of the experiments shown in Figure 4 should be provided.

Reviewer #3 (Remarks to the Author):

Authors describe the screening of the 13, 500 compounds from the GlaxoSmithKline Tres Cantos Antimalarial Set (TCAMS) against stage V gametocytes in the *P. falciparum* Female Gametocyte Activation Assay (Pf FGAA). The objective of this study was to identify new chemical entities with activity against both asexual blood stages (previous work by GSK scientists) and stage V (female) gametocytes (this manuscript). This led to the identification of compounds effective against female gamete formation. Active compounds were biologically profiled not just for activity against female gametocytes but also in terms of their speed of action.

Although authors claim that it is the first time compounds effective against gamete formation have specifically been identified, the concept of screening chemical libraries to identify compounds that target late stage gametocytes is not new. Indeed several reports of primary screens of chemical libraries for compounds with late stage gametocytocidal activity exist in the literature.

Although it is clear that identified compounds target stage V (female) gametocytes, the possibility that some of the compounds can target early stage gametocytes cannot be completely ruled out. Authors need to provide data in this regard to rule out this possibility. This is especially important for compounds determined to be slow acting in the speed of action assay. A slow acting compound on early gametocytes could simply indicate preference for more mature, late gametocyte stages. It is therefore important to be able to confirm stage-specificity especially for the slow acting compounds. The ability of a compound to also target early stage gametocytes has recently received attention in view of the hypothesis that a compound targeting asexual parasites, with the added ability to target early gametocyte development is predicted to increase the transmission-blocking potential by decreasing the gametocyte load. Therefore, stage-specificity of action of compounds against both immature and mature gametocytes should be verified and/or confirmed.

The novelty of the work presented in this manuscript lies mainly in using an assay that utilizes Plasmodium gamete formation as the endpoint. However, whether or not active

compounds identified from this assay would have been undetected by previous assays, this alone does not provide a significant advance warranting publication in Nature Communications. This is in view of the fact that the parasite lifecycle may be interrupted by solely targeting one of either female or male gametes since both are required for the development of mosquito stages. The various primary screens of chemical libraries reported in literature for compounds with late stage gametocytocidal activity ought to have been able to detect compounds which inherently target female and/or male gametes.

Reviewer #1 (Remarks to the Author):

There is an urgent need for additional antimalarials against all stages of the parasite, including the transmission stages and this manuscript identifies 3 scaffolds that when incubated *in vitro* at micromolar concentrations with late stage sexual stages reduce transmission in a standard membrane feed. This is encouraging, but statistics and the actual numbers of oocysts and infected mosquitoes need to be included to understand the significance of the SMFA experiment.

RE: A detailed table with all this information about the SMFA experiments has been included as supplementary data (Supplementary Table 4).

For these compounds the term "transmission blocking" also needs to be changed to "transmission reducing" throughout the text.

RE: We disagree with the reviewer. The term "potential" has been also included along with "transmission-blocking" when we speak about the compounds tested, therefore we do not consider necessary to change it by "transmission-reducing".

In addition for publication in Nature Communications they need to test for *in vivo* efficacy using the rodent malaria model or indicate that the compounds were toxic to mice and could not be tested. This *in vivo* data is needed to get some sense of how much additional drug development is needed to advance these compounds and must be included.

RE: We fully agree with the reviewer. *In vivo* efficacy data would shed light on the developability of these compounds as antimalarial drugs. Two out of the six compounds tested in the SMFA were further studied in a *Plasmodium berghei* mouse model (see the corresponding Results and Discussion sections). The compounds identified in this paper are promising screening hits. Further chemistry efforts are required to improved their parasitological potency, physicochemical and DMPK properties in order to select pre-clinical candidates

To direct further screening efforts, they should also directly compare SMFA results from randomly selected compounds shown in figure 2 to inhibit gametocyte viability and those that do not.

RE: All the compounds in Figure 2 inhibit gametocyte viability in terms of gamete formation but only 30% of them also inhibited gametocyte metabolism fast enough to be detected in the ATP assay. We decided to prioritise those compounds exclusively detected by the *Pf* FGAA because they represent new transmission-blocking chemical diversity. Those detected also in the ATP-assay were previously studied in the TCAMS screen described in Almela *et al.* 2015, where some randomly compounds were also tested in the SMFA.

Gametocyte assays are not easy and it is important to know if it would be more efficient to screen first for viability or *Pf* FGAA. Their finding that only 4 of the 6 selected compounds from a total of 405 identified with the *Pf* FGAA assay alone may reduce transmission at micromolar concentrations is not a strong endorsement of their *Pf* FGAA only screening strategy.

RE: Based on the results obtained in the present study, the *Pf* FGAA has been proven to identify chemical entities that the ATP-assay was not able, mainly due to the biological processes that each of the assays cover, i.e. metabolic vs. gamete formation pathways. Therefore, the use of the ATP assay alone may reduce the number of identified targets when screening a collection of compounds, hence miss potential useful transmission-blocking compounds. Our results suggest that the use of more than one assay may help to identify a higher diversity of biological targets. Regarding to the SMFA results, 4 out of the 6 compounds tested showed a reduction in transmission but this proportion might vary if the number of hits tested were increased.

Line 69 & 228: Please remove “increased” and revise the sentence. These assays focus on a distinct stage, but this does not mean that they have more biological content than previously used gametocyte screens that monitored ATP levels or mitochondrial potential or redox activity or luciferase expression.

RE: The term “increased” is used here because the current assay measures the inhibitory activity of compounds from stage V gametocytes until female gamete, whereas previous gametocyte assays endpoint was just late stage gametocytes (stages IV and V). Therefore, the biology of the parasite that is covered with this assay is increased because, as the reviewer mentioned, it focuses on a different stage although the starting point is the same. Consequently, the number of potential targets has been increased notably. For a better understanding, the term “increased” has been replaced by “extended” throughout the manuscript.

Line 70: As described above data needs to be presented to state that “the in vitro gamete formation assays best approximates the activity of drugs in the mosquito stages, as determined by the ex vivo assay.” No reference or data is given for this statement.

RE: Regarding to the parasite cell biology involved, gamete formation assays are the gametocyte assays that better overlap SMFA to date (see Figure 1 in Ruecker *et al.* 2014 below) because the endpoint of these assays is a process that naturally occurs in the mosquito. The corresponding reference have been included (new line 75).

[redacted]

Line 140: Which 12 compounds blocked Pfs25 expression when administered after the emergence stim? Please include this in a chart and indicate them on fig 2, perhaps by circling them or using a different color star.

RE: We have included an additional column in Supplementary Table 3 with the speed of action classification (fast, slow and gamete). Besides, the 12 compounds that showed activity without previous exposure to gametocytes have been highlighted in Figure 2 (purple stars).

Line 147: In figure 2 please identify the final 6 compounds evaluated in detail as described above. What was the effect of these compounds on viability?

RE: The 6 compounds have been highlighted in Figure 2 (pink stars). These compounds were considered inactive in the ATP-assay as they showed values of inhibition below 40% at 2 μ M concentration (Supplementary Table 3). We have included these data in Table 2 for a better biological profile overview.

Line 147: More specifics are needed about how the 6 compounds were selected. Were other compounds tested and not found to be effective in the SMFA? Those selected did not include all the most potent compounds and only 4 clusters were tested, so diversity and potency were not the defining priorities. Clusters 3, 13, 26, 57 and singleton 1 all had equally potent compounds.

RE: Prioritization of clusters and final selection for SMFA test was defined after an extensive data mining in the GSK database. Chemical series that were previously studied in medicinal chemistry programs were discarded along with those possessing a toxic *in silico* phys-chem profile. Subsequently, four distinct clusters showing diverse and promising biological profiles were prioritised. Potency (IC₅₀) and cytotoxicity (TOX₅₀) of the compounds of each of these clusters were finally taken into account for progression to SMFA.

Line 148: Please replace “a diversity of” with “4 distinct” chemical clusters were represented.

RE: It has been replaced (new line 153).

Line 160: The number of oocysts and infected mosquitoes observed in the control and after compound treatment need to be included. If the mean for the control was 2 oocysts in 70% of the mosquitoes, which are the minimums indicated in the methods (line 316), it is difficult to know the significance of 93% inhibition. Statistics are needed to confirm that these compounds significantly reduce transmission. The concentrations used for the SMFA need to be included in Table 2.

RE: A complete dataset of experimental raw data, including statistical analysis, has been included as supplementary material (Supplementary Table 4 and new line 339). Concentrations used for SMFA were IC₉₀ (μM) obtained in the *Pf* FGAA using 48h incubation of stage V gametocytes (lines 158). To clarify, “IC₉₀” has been replaced for “SMFA-concentration tested” in Table 2.

Lines 173 & 174: Please add the cluster numbers for these two scaffolds and their EC₅₀s in the text. Were these compounds tested in the time course? Was there a reason they were not tested in the SMFA assay?

RE: Cluster numbers and IC₅₀s have been included in the text (new lines 199 & 200). Both compounds were tested in the time course, where TCMDC-138933 showed a slow-acting profile while TCMDC-137820 showed fast-acting (Supplementary Table 3). None of these compounds were initially considered for SMFA because their selectivity ratios were < 10 (Supplementary Table 1).

Line 204: Please clarify this sentence. I thought all the compounds were effective against asexual parasites. At the least the compounds listed in table 1 have activity against asexual parasites. Is the sentence referring to the lack of an effect for most of the compounds in the viability assay? Although interesting from a biological perspective, from the drug development side it is a concern that the selected compounds do not affect gametocyte viability (Fig 2).

RE: All the compounds contained in the TCAMS collection are active against asexual blood stages with potencies below 2μM (line 82). The sentence in line 225 & 226 means that 90%

of the compounds further profiled were active against stage V gametocytes but not against mosquito stages, i.e. female gametes, as they did not show activity ($IC_{50} > 2\mu M$) when the parasites were exposed to the drug after gamete formation.

The 90 prioritised compounds showed activities below 40% in the ATP gametocyte assay but this only means that they were undetectable in this metabolic assay under the conditions tested. Since there is not gamete formation after compound treatment, gametocyte viability is clearly affected. These 90 compounds might be hitting targets that are not detected using an ATP depletion assay. This fact highlights the value of the *Pf* FGAA, since this assay identified hits that are not detected using metabolic assays.

What stages are not affected by the compounds? Will drug levels have to be maintained until all the sequestered stage I-IV gametocytes mature to stage V? The discussion needs to more clearly address this issue.

RE: The main objective of our study was the identification of molecules with activity against both asexual blood stages and stage V gametocytes for a single daily dose during 3 days of treatment. Ideally, drug levels might need to maintain until maturation of all stages, otherwise the drug might need several administrations to kill early stages as long as they mature.

Line 225: Please include more discussion about why activity against females is not sufficient to predict transmission-blocking activity and how this impacts utility of this type of screen.

This sentence was misleading and has been removed from the manuscript.

Since zygote formation and onwards development of mosquito stages require both female and male functional gametes, the parasite lifecycle would be theoretically interrupted by solely targeting one of them. However, male gamete formation has been described to be a more sensitive process than female formation and several drugs have shown to be active against exflagellation with no effect in female gamete formation. Consequently, the screening of libraries targeting also the male gamete formation process will definitely increase the number of potential starting points to discover new transmission-blocking drugs. Based on this fact, our next step will be to scale-up the *P. falciparum* Dual Gamete Formation Assay (*Pf* DGFA) to high throughput format, providing therefore a closer *in vitro* approach to the SMFA suitable for HTS of large compound libraries.

Reviewer #2 (Remarks to the Author):

This is concise, clear, and focused manuscript, easy to read. Authors describe logical and efficient step-wise approach to identify compounds inhibiting development of gametocytes, provide 6 well-characterized *in vitro* and *ex vivo* inhibitory compounds, and a long list of active compounds for future characterization. The manuscript will be interesting to malaria researchers and identified compounds are a good starting point in the development of transmission blocking drugs.

Nevertheless, this reviewer has a few minor suggestions/questions.

1. Authors should state in Figures 5 and 6 what their error bars are (I assume these are Standard Deviations).

RE: Error bars represent standard deviations. This information has been included in the corresponding figures.

2. This reviewer did not understand why only compounds with <40% inhibition in ATP assay went to further biological characterization.

RE: We decided to prioritise for progression the compounds that were only detected by this functional assay as they supposed chemical novelty. Most of the compounds that showed activities >40% in the ATP assay were previously studied by Almela and colleagues in a previous TCAMS screen (Almela *et al.* 2015).

3. Figure 4 states specific numbers for the inhibitory compounds in each group (slow, fast, and gamete). If possible, it would be nice to indicate in the list of 94 compounds which of them belong to which group. Also, in the scheme it says that “Minimum incubation time required for activity ($IC_{50} < 2 \mu M$)”. I guess, it means that all compounds with $IC_{50} < 2 \mu M$ (measured previously at 48 hours incubation) were tested in this experiment using 24 and 0 time incubation but at what concentration? Was it a single 2 μM concentration used and efficiency of inhibition determined by the appearance of gamete? In the Supplementary table 3 IC_{50} values for 24 and 0 time points are shown – it means that dose-response data were also obtained at these times, but this is not described in the manuscript.

RE: Information about the Pf FGAA classification (slow, fast and gamete) have been included in Supplementary Table 3, where IC_{50} values are shown. The 90 compounds were tested in dose-response (line 137) and we considered “active” those with IC_{50} values below 2 μM .

This reviewer counted in the Supplementary Table 3 twelve compounds with $IC_{50} < 3 \mu M$ in PfGAA at time “0” belonging to 6 groups. Are these the compounds that were active in gamete-targeting assay (two compounds indicated in the Fig 3 gamete targeting group are among these 12 that reviewer has found)? If so, it would be interesting to discuss that they might be active at time “0” because they have the lowest IC_{50} at time “0” and the experiment in this setup provides specific IC_{50} threshold for selection of active compounds. In any case, a better description of the experiments shown in Figure 4 should be provided.

RE: The 12 compounds classified as gamete-targeted showed activity when were added after triggering gametocyte activation (i.e. time “0”). We considered a cut off of 2 μM to define compound activity at any of the time points used in the speed of action experiments. As required by the reviewer, a better description have been included in line 145.

Reviewer #3 (Remarks to the Author):

Authors describe the screening of the 13, 500 compounds from the GlaxoSmithKline Tres Cantos Antimalarial Set (TCAMS) against stage V gametocytes in the *P. falciparum* Female Gametocyte Activation Assay (Pf FGAA). The objective of this study was to identify new chemical entities with activity against both asexual blood stages (previous work by GSK scientists) and stage V (female) gametocytes (this manuscript). This led to the identification of compounds effective against female gamete formation. Active compounds were biologically profiled not just for activity against female gametocytes but also in terms of their speed of action.

Although authors claim that it is the first time compounds effective against gamete formation have specifically been identified, the concept of screening chemical libraries to identify compounds that target late stage gametocytes is not new. Indeed several reports of primary screens of chemical libraries for compounds with late stage gametocytocidal activity exist in the literature.

RE: The novelty of this screen is not the evaluation of chemical libraries against stage V gametocytes but the use of gamete formation as endpoint of the assay with the aim to assess functional damage that previous gametocyte assays were not able to evaluate.

Although it is clear that identified compounds target stage V (female) gametocytes, the possibility that some of the compounds can target early stage gametocytes cannot be completely ruled out. Authors need to provide data in this regard to rule out this possibility.

RE: We agree with the reviewer about the possibility that some compounds might have additional activity in early stage gametocytes. However, the assay described in the manuscript does not study early stage gametocyte viability and is specifically designed to exclude them as a potential confounding factor. Indeed, such activity is not the focus of the study nor particularly relevant to interpretation of our data.

This is especially important for compounds determined to be slow acting in the speed of action assay. A slow acting compound on early gametocytes could simply indicate preference for more mature, late gametocyte stages. It is therefore important to be able to confirm stage-specificity especially for the slow acting compounds. The ability of a compound to also target early stage gametocytes has recently received attention in view of the hypothesis that a compound targeting asexual parasites, with the added ability to target early gametocyte development is predicted to increase the transmission-blocking potential by decreasing the gametocyte load. Therefore, stage-specificity of action of compounds against both immature and mature gametocytes should be verified and/or confirmed.

RE: The speed-of-action experiments were all performed using mature stage-V gametocytes, so what the reviewer suggests about the preference of slow compounds for early stages is therefore unlikely. These authors consider that compounds with activity against the transmissible circulating stages – stage-V gametocytes - are more predictive of transmission interruption than sequestered immature stages.

The novelty of the work presented in this manuscript lies mainly in using an assay that utilizes Plasmodium gamete formation as the endpoint. However, whether or not active compounds identified from this assay would have been undetected by previous assays, this alone does not provide a significant advance warranting publication in Nature Communications. This is in view of the fact that the parasite lifecycle may be interrupted by solely targeting one of either female or male gametes since both are required for the development of mosquito stages. The various primary screens of chemical libraries reported in literature for compounds with late stage gametocytocidal activity ought to have been able to detect compounds which inherently target female and/or male gametes.

RE: We agree with the reviewer about the potential identification in previous screens of compounds with activity against male and/or female gametocytes and, therefore, their relative gametes. Nevertheless, as we have proven in this study, previous gametocyte screens were not able to identify compounds that affect the gamete formation since they measure a direct effect on the mature gametocyte. Consequently, the comparison between ATP-assay (as an example of previous late stage gametocyte assays) and *Pf* FGAA identifies an important chemical diversity involved in the gamete formation process that previous screens were not able to identify, proving the value of this new assay for the scientific community.

REVIEWERS' COMMENTS:

Reviewer #1 (Remarks to the Author):

In response to the prior critiques the authors added in vivo efficacy data for 2 compounds. Unfortunately, both compounds tested were metabolized very quickly and did not block asexual parasite growth so were not further tested for transmission.

The data presented comparing previously reported ATP/viability results and the PFGA time course assays is useful for further basic research on gametocyte and gamete development although this manuscript does not focus on the mechanisms of action of the compounds. They do show that compounds from 3 of the 5 scaffolds that were identified by the PFGA, but not the ATP assay, do block transmission in an SMFA. Interestingly, yet not highlighted in the manuscript the SMFA data suggest that contrary to their initial PFGA premise that only one sexual stage needs to be eliminated, it seems that both male exflagellation and female gamete formation need to be inhibited to effectively block transmission in an SMFA.

From the results presented, I'm not convinced that the PFGA only compounds would be top on my list for further drug development, but the authors are correct that the assay does identify compounds that have an in vitro effect on gametes, which were not identified by the ATP assay. The jury is still out on whether this can be translated to in vivo drug development. It is very difficult to block transmission and focusing on compounds that may not be cidal is a concern.

Specific comments

Abstract Line 32: The need to clearly indicate that the in vitro transmission blocking activity was only confirmed (>80%) in 4 of the 6 compounds tested and these were from just 3 distinct scaffold clusters.

46: replace ideal with important. Eliminating stage Vs alone is not going to do much, unless the drug is chronically administered to kill the Vs as they continue to be produced by other stages.

75: Please remove this claim, until you have directly compared compounds that do and do not affect gametocyte viability in an SMFA. Reference 25 only tests three malaria box compounds with activity against gametes in an SMFA. None of the malaria box compounds have been found to consistently, and potently block gametocyte viability. Compounds that block gamete formation should work in an SMFA. It is more of a surprise that some of the compounds do not work.

196: Need to indicate how many of these are cytotoxic for HepG2

227: As indicated before this sentence needs to be clarified. Please clarify the sentence 90% exert their effect exclusively in stage V gametocytes (as asexual blood stages) Do you mean that they do not kill gametes or that they are effective against early stage gametocytes, sporozoites or liver stages?

229: Please explain why the gamete active compounds still need to be tested for activity against stage V gametocytes? You only selected ones that were negative against gcytes in the ATP assay. Do you mean giving a pulse then washing off before stimulating gametogenesis.

237: Please add that this suggests that compounds that affect both males and females are more efficacious.

254: Please indicate the actual % that were in the optimal physicochemical space.

Table S4: Please add a line indicating the Prevalence of infection in test compound.

Also in exp 2 for 124559 need to indicate with a footnote that the significant p value is for an increase in oocysts not a decrease.

Reviewer #2 (Remarks to the Author):

Authors answered all points raised by this reviewer satisfactory. The title of section in line 167 is misleading in view of this reviewer, as it does not test for the blocking of gametocytogenesis of *P. falciparum*. Should they inhibit *B. bergeri* growth at all and what relevance it has to the anti-gemtocytogenesis activity identified here? A better title would be something like "Insights into pharmacokinetic studies using in vivo mouse model"

Reviewer #1 (Remarks to the Author):

Abstract Line 32: *The need to clearly indicate that the in vitro transmission blocking activity was only confirmed (>80%) in 4 of the 6 compounds tested and these were from just 3 distinct scaffold clusters.*

RE: Following the reviewer's suggestion, it has been indicated in the abstract (new line 34).

Line 46: *Replace ideal with important. Eliminating stage Vs alone is not going to do much, unless the drug is chronically administered to kill the Vs as they continue to be produced by other stages.*

RE: The qualifier "ideal" has been replaced by "important" (new line 48).

Line 75: *Please remove this claim, until you have directly compared compounds that do and do not affect gametocyte viability in an SMFA. Reference 25 only tests three malaria box compounds with activity against gametes in an SMFA. None of the malaria box compounds have been found to consistently, and potently block gametocyte viability. Compounds that block gamete formation should work in an SMFA. It is more of a surprise that some of the compounds do not work.*

RE: This paragraph has been rewritten for clarification (new lines 71-79).

Line 196: *Need to indicate how many of these are cytotoxic for HepG2 .*

RE: The percentage of molecules with selectivity index (SI) >10 has been indicated (new line 196).

Line 227: *As indicated before this sentence needs to be clarified. Please clarify the sentence 90% exert their effect exclusively in stage V gametocytes (as asexual blood stages) Do you mean that they do not kill gametes or that they are effective against early stage gametocytes, sporozoites or liver stages?*

RE: The sentence has been clarified (new lines 227-229). We have proven that 90% of the compounds profiled in this study exerted their activity in stage V gametocytes (and asexual blood stages) but not in the female gamete because they were inactive (IC₅₀ < 2μM) when compounds were added after triggering gametocyte activation. The compounds have not been further tested in other *P. falciparum* stages such as early-stage gametocytes or liver stages.

Line 229: *Please explain why the gamete active compounds still need to be tested for activity against stage V gametocytes? You only selected ones that were negative against gcytes in the ATP assay. Do you mean giving a pulse then washing off before stimulating gametogenesis.*

RE: These compounds could have some additional activity over stage V gametocytes, apart from the already known activity against female gametes. To confirm stage-specificity, we

would need to perform additional experiments washing out the drug before proceeding to gametocyte activation. This method is described in Ruecker *et al.* 2014 (reference 25 in the manuscript).

Line 237: *Please add that this suggests that compounds that affect both males and females are more efficacious.*

RE: We have included the following sentence in lines 240 & 241: "This suggests that compounds affecting both male and female gametes may lead to a more efficacious blockade of malaria transmission".

Line 254: *Please indicate the actual% that were in the optimal physicochemical space.*

RE: It has been indicated (new line 258).

Table S4: *Please add a line indicating the Prevalence of infection in test compound. Also in exp 2 for 124559 need to indicate with a footnote that the significant p value is for an increase in oocysts not a decrease.*

RE: They have been indicated in Supplementary Table 1 (previous Table S4).

Additional requests:

*"To more clearly indicate this focus on late stages they should revise the **line 29** in the abstract by adding the qualifier "late stage" gametocyte-targeting.*

RE: It has been indicated (new line 31).

*The sentence (**line 226-228**) that indicates "Overall, almost 90% of the compounds tested exerted their effect exclusively in stage V gametocytes (and asexual blood stages) needs to be qualified because they have not established that the effects are exclusively in stage Vs (and asexual blood stages)."*

RE: It has been clarified (lines 228-230) and the word "exclusively" has been removed.

Reviewer #2 (Remarks to the Author):

*Authors answered all points raised by this reviewer satisfactory. The title of section in line 167 is misleading in view of this reviewer, as it does not test for the blocking of gametocytogenesis of *P. falciparum*. Should they inhibit *P. berghei* growth at all and what relevance it has to the anti-gametocytogenesis activity identified here? A better title would be something like "Insights into pharmacokinetic studies using in vivo mouse model"*

RE: The *in vivo* mouse model used here evaluates the efficacy of compounds against *P. berghei* asexual blood stages. The two compounds tested were active (IC50<1uM) against these stages *in vitro* but they didn't show a reduction of parasitemia *in vivo*. Based on the results

obtained in the additional *in vivo* pharmacokinetic and *in vitro* ADME studies, the lack of efficacy in the mouse model could be due to a low bioavailability profile of these two molecules.

For a better understanding, the title of this section has been replaced by “*In vivo* efficacy and pharmacokinetics of selected molecules” (new line 171).